# Phosphopeptide binding by Sld3 links Dbf4-dependent kinase to MCM replicative helicase activation

Tom D Deegan[†], Joseph TP Yeeles & John FX Diffley[*]

## Abstract

The initiation of eukaryotic DNA replication requires the assembly of active CMG (Cdc45-MCM-GINS) helicases at replication origins by a set of conserved and essential firing factors. This process is controlled during the cell cycle by cyclin-dependent kinase (CDK) and Dbf4-dependent kinase (DDK), and in response to DNA damage by the checkpoint kinase Rad53/Chk1. Here we show that Sld3, previously shown to be an essential CDK and Rad53 substrate, is recruited to the inactive MCM double hexamer in a DDK-dependent manner. Sld3 binds specifically to DDK-phosphorylated peptides from two MCM subunits (Mcm4, 6) and then recruits Cdc45. MCM mutants that cannot bind Sld3 or Sld3 mutants that cannot bind phospho-MCM or Cdc45 do not support replication. Moreover, phosphomimicking mutants in Mcm4 and Mcm6 bind Sld3 without DDK and facilitate DDK-independent replication. Thus, Sld3 is an essential "reader" of DDK phosphorylation, integrating signals from three distinct protein kinase pathways to coordinate DNA replication during S phase.

Keywords   DDK; DNA replication initiation; Sld3
Subject Categories   DNA Replication, Repair & Recombination
The EMBO Journal (2016) 35: 961–973

See also: **H Araki** (May 2016)

## Introduction

To ensure that the genomes of eukaryotic cells are completely replicated once during each cell cycle, the motor of the replicative helicase, MCM, is tightly regulated during the initiation of chromosome replication. MCM is first loaded at DNA replication origins as an inactive double hexamer during G1 phase of the cell cycle (Evrin *et al*, 2009; Remus *et al*, 2009). Dbf4-dependent kinase (DDK) and cyclin-dependent kinase (CDK) subsequently activate MCM during S phase by promoting assembly of a stable CMG (Cdc45-MCM-GINS) helicase in a reaction requiring numerous essential firing factors

(Siddiqui *et al*, 2013; Tanaka & Araki, 2013). Whilst the mechanism of MCM activation remains unclear, the recent reconstitution of origin firing with purified proteins has identified the minimal set of firing factors required for this reaction: CDK, DDK, Sld3/7, Cdc45, Sld2, Dpb11, DNA polymerase ε, GINS and Mcm10 (Yeeles *et al*, 2015).

Budding yeast Sld3 and its human orthologue Treslin/TICRR are emerging as key firing factors that are regulated by S-phase kinase signalling. CDK phosphorylation of Sld3 and Treslin/TICRR is essential for replication (Tanaka *et al*, 2007; Zegerman & Diffley, 2007; Boos *et al*, 2011; Kumagai *et al*, 2011), and the DNA damage checkpoint kinases (Rad53 in budding yeast, Chk1 in humans) inhibit replication initiation in part through Sld3 and Treslin/TICRR (Lopez-Mosqueda *et al*, 2010; Zegerman & Diffley, 2010; Boos *et al*, 2011; Guo *et al*, 2015).

Recent evidence suggests that Sld3 along with Sld7 and Cdc45 may function in the DDK-dependent regulation of origin firing. A number of studies have established that DDK phosphorylates MCM on its Mcm2, Mcm4 and Mcm6 subunits (Labib, 2010) and DDK promotes the recruitment of Sld3/7 and Cdc45 to replication origins (Heller *et al*, 2011; Tanaka *et al*, 2011a; Yeeles *et al*, 2015), which can occur in the absence of CDK (Heller *et al*, 2011; Yeeles *et al*, 2015). A deletion in the N-terminal domain of Mcm4 can bypass the requirement for DDK *in vivo* (Sheu & Stillman, 2010), suggesting that MCM phosphorylation counteracts an inhibitory activity intrinsic to this region. However, a point mutation in Mcm5 can also bypass the requirement for DDK (Hardy *et al*, 1997). Thus, it is entirely unclear how DDK promotes recruitment of Sld3/7 and Cdc45 to MCM and how Sld3/7 and Cdc45 contribute to each other's recruitment. Moreover, it is unclear whether MCM phosphorylation by DDK plays any additional roles in promoting replication (Bruck & Kaplan, 2015). Here we describe the molecular mechanism by which DDK mediates this first step in helicase activation using purified budding yeast proteins.

## Results

### DDK promotes Sld3-dependent Cdc45 recruitment

To investigate the mechanism of DDK-dependent Sld3/7 and Cdc45 recruitment, we utilised a recently developed *in vitro* system, which

The Francis Crick Institute, Clare Hall Laboratory, South Mimms, Herts, UK
  *Corresponding author. Tel: +44 1707625869; E-mail: john.diffley@crick.ac.uk
  [†]Present address: The MRC Protein Phosphorylation and Ubiquitylation Unit, The Sir James Black Centre, College of Life Sciences, University of Dundee, Dundee, UK

recapitulates DNA replication origin firing with purified budding yeast proteins (Yeeles *et al*, 2015). In this system, MCM, comprising six related subunits (Mcm2-7), is first loaded onto DNA attached to paramagnetic beads using purified origin recognition complex (ORC), Cdc6 and MCM•Cdt1. The loaded MCM is phosphorylated with DDK and incubated with purified Cdc45 and Sld3/Sld7. Addition of CDK, Sld2, DNA polymerase ε, Dpb11, GINS and Mcm10 then generates complexes containing all of the firing factors. Finally, DNA synthesis occurs upon addition of DNA polymerase α, RPA, Ctf4 and topoisomerase II. When CDK is omitted, Cdc45 and Sld3/7, but none of the other firing factors, are recruited to MCM (Yeeles *et al*, 2015). Consistent with this, Fig 1A and B shows that Sld3/7 and Cdc45 were recruited to MCM in an ORC- and DDK-dependent manner in the absence of all other firing factors. Recruitment of Cdc45 required Sld3/7, but recruitment of Sld3/7 did not require Cdc45 (Fig 1B, lanes 4 and 5). Moreover, Sld3/7 was recruited efficiently even after MCM loading factors were removed by high salt extraction (Fig 1C). Therefore, DDK promotes Sld3/7 binding to MCM, which in turn recruits Cdc45. The association of Cdc45 and Sld3 to origins *in vivo* by chromatin immunoprecipitation (ChIP) was previously reported to be interdependent in budding yeast (Kamimura *et al*, 2001); it may be that Sld3 becomes unstable *in vivo* in the absence of Cdc45, which would explain the apparent interdependence in such experiments. Alternatively, our conditions *in vitro* may not be stringent enough to detect any role for Cdc45 in Sld3 recruitment. Nonetheless, we note that our results are consistent with results in fission yeast, in which Sld3 origin association clearly precedes Cdc45 (Nakajima & Masukata, 2002; Yabuuchi *et al*, 2006).

Figure 1D (lanes 3, 4) shows that Sld3 can recruit Cdc45 to the MCM double hexamer without Sld7. An unstructured loop in the conserved Sld3–Treslin domain (STD, Fig 1E) has been shown to bind Cdc45 (Itou *et al*, 2014). We generated several clusters of point mutants in this region (Fig 1E) and found some which were defective in Cdc45 binding (Fig 1F). We characterised further Sld3-3E1, which has the fewest mutations. Figure 1D shows that this mutant still bound MCM in a DDK-dependent manner, but was defective in Cdc45 recruitment to the double hexamer. The Sld3-3E1 mutant was also unable to support DNA replication (Fig 1G and H) indicating that Cdc45 binding by Sld3 is required for Cdc45 recruitment to MCM and subsequent DNA replication.

## MCM binding is an essential function of Sld3

We next sought to find the MCM-binding activity in Sld3/7. Sld3 and Sld7 can each independently bind loaded MCM in a DDK-dependent manner (Fig 2A), contrary to previously reported data showing the association of Sld7 with replication origins *in vivo* to be dependent on Sld3 (Tanaka *et al*, 2011b). Increasing the salt concentration in our *in vitro* experiments abolished the ability of Sld7, but not Sld3, to bind directly to MCM (Fig EV1A). Together with previously published ChIP experiments (Tanaka *et al*, 2011b), we suggest this weak Sld7–MCM interaction plays no significant role in Sld3 recruitment to MCM *in vivo*.

Whilst Sld3 is required for DNA replication *in vitro*, Sld7 is not (Fig 2B) consistent with the fact that Sld3, but not Sld7, is essential for viability (Kamimura *et al*, 2001; Tanaka *et al*, 2011b). We therefore focussed on Sld3. From analysis of deletion mutants (Figs 2C

and EV1), we identified a region of Sld3 critical for MCM binding that lies between the STD and the essential CDK sites (T600, S622) (Fig 2D). Many of the Rad53 phosphorylation sites that contribute to the inhibition of origin firing in response to DNA damage lie within and around this region required for MCM interaction (Zegerman & Diffley, 2010). As shown in Fig 2E, incubation of Sld3 with Rad53 blocked the DDK-dependent interaction of Sld3 with MCM in an ATP-dependent manner. Together with previous work showing that Rad53 blocks Sld3's ability to interact with both Dpb11 and Cdc45 (Zegerman & Diffley, 2010), this contributes to the idea that Rad53 phosphorylation inactivates Sld3 by blocking multiple, critical protein–protein interactions.

We next generated a series of point mutations in this region of Sld3 and found several which showed reduced DDK-dependent MCM binding (Fig EV2). We combined these to generate the Sld3-6E mutant in which six conserved basic residues were changed to glutamate (Fig 2D). This mutant exhibited normal Cdc45 and Sld7 binding (Fig EV2C) but was defective in DDK-dependent MCM binding (Fig 2F).

We next assessed the effect of the MCM-binding mutations in Sld3 on DNA replication. In the presence of the complete set of firing factors (Fig 3A), the recruitment defect of Sld3-6E was partially rescued (compare Figs 2F and 3B). This residual recruitment of the Sld3-6E mutant protein (Fig 3B and C) required Sld7 (Fig 3C, compare lanes 4 and 8), consistent with the fact that Sld7 alone can bind MCM in a DDK-dependent manner in the absence of Sld3 (Fig 2A). However, even in the presence of Sld7, the Sld3-6E mutant protein was unable to support GINS recruitment (Fig 3B) or DNA replication *in vitro* (Fig 3D). Furthermore, haploid yeast cells harbouring the *sld3-6E* mutant allele at the endogenous *SLD3* locus were unable to grow, indicating that *sld3-6E* cannot support viability *in vivo* (Fig 3E). Together, these results indicate that the ability of Sld3 to bind directly to DDK-phosphorylated MCM and recruit Cdc45 is essential for DNA replication. Moreover, recruitment of Sld3 to MCM via Sld7 cannot support CMG assembly or replication.

## Sld3 interacts with phosphorylated Mcm4 and Mcm6

We sought to identify DDK-dependent Sld3-binding sites within the MCM double hexamer. This was complex because DDK phosphorylates multiple MCM subunits on multiple sites (Randell *et al*, 2010) and because DDK only phosphorylate MCM efficiently in the context of the loaded double hexamer (Francis *et al*, 2009). Therefore, we needed to devise an approach to separate MCM subunits after phosphorylation. To accomplish this, we utilised the fact that MCM can "slide" off the end of a linear DNA molecule in high salt (Remus *et al*, 2009). We phosphorylated the loaded double hexamer, allowed it to slide off at high salt and observed that prolonged incubation under these conditions was sufficient to disrupt interactions within the MCM ring (Fig EV3) consistent with previous work showing MCM complexes are unstable in high chloride salt when not bound to DNA (Adachi *et al*, 1997). By using Sld3/7 attached to beads via a FLAG tag on Sld3, we were thus able to determine which individual subunits Sld3/7 bound to (Fig 4A). Figure 4B shows that Sld3/7 bound only two subunits, Mcm4 and 6, in a DDK-dependent manner. The binding to Mcm4 and 6 was dependent upon DDK and Sld3/7 and was sensitive to phosphatase treatment (Fig 4C). Wild-type Sld3 without Sld7 also bound both

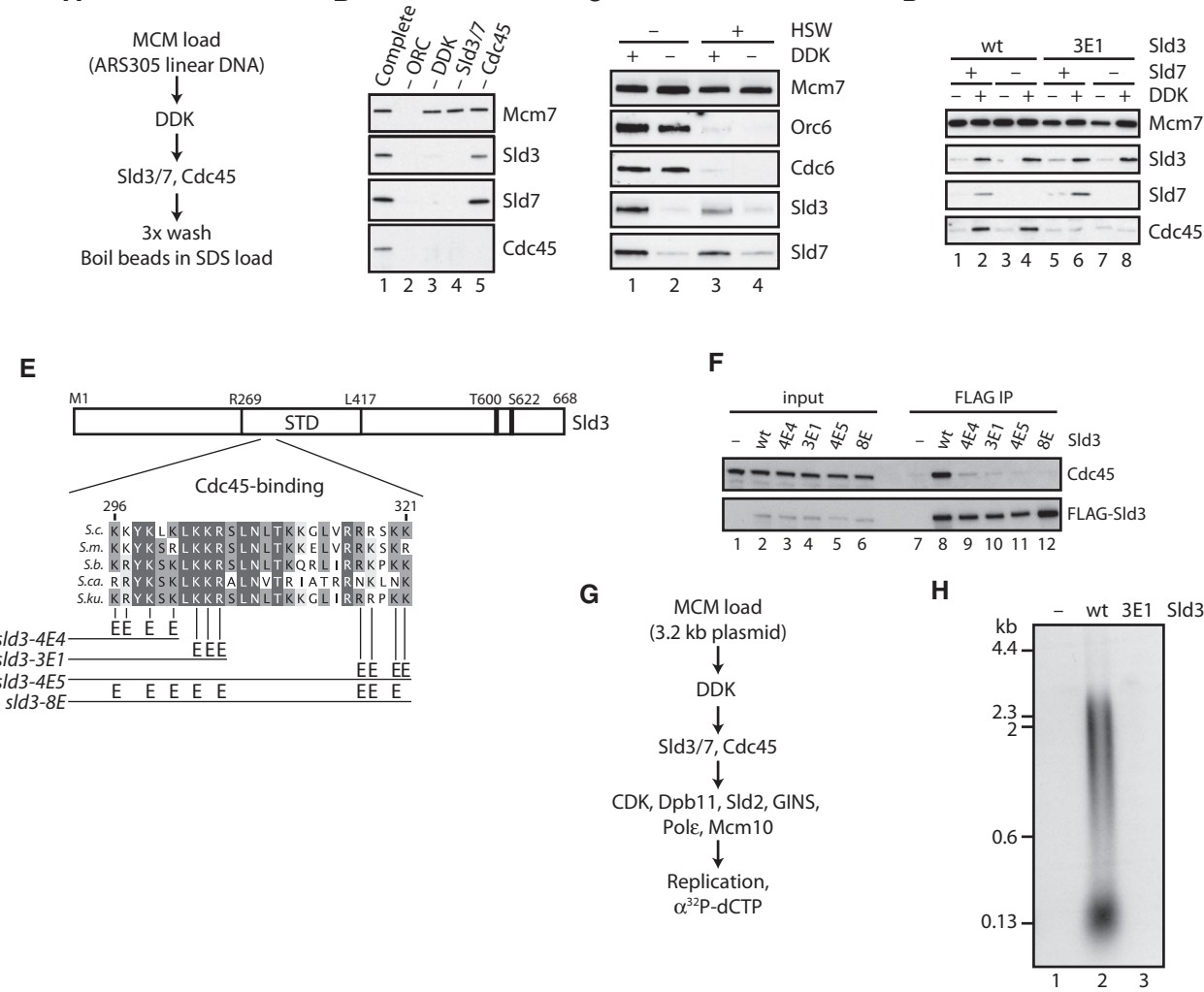

**Figure 1.  DDK promotes Sld3-dependent Cdc45 recruitment.**

A    Reaction scheme for Sld3/7/Cdc45 recruitment experiments. DNA substrate was bound to beads, and reaction mixtures were removed after each step.

B, C  Immunoblots of recruitment reactions performed as described in (A), with the indicated proteins omitted. In (C), a mid-reaction wash in high salt (0.5 M NaCl) buffer (HSW) was included following DDK phosphorylation as indicated.

D    Recruitment reaction performed as in (A), using wild-type (wt) or a Cdc45-binding mutant (3E1) of Sld3.

E    Schematic showing the position of Cdc45-interacting region in Sld3. Alignments were generated in Jalview using Clustal. Sld3 from various fungal species is included (S.c., *Saccharomyces cerevisiae*, S.m., *Saccharomyces mikatae*, S.b., *Saccharomyces bayanus*, S.ca., *Saccharomyces castellii*, S.ku., *Saccharomyces kudriauzevii*). Residue numbers correspond to *S. cerevisiae* Sld3. Residues were substituted for Glu as indicated.

F    Wild-type or mutant FLAG-Sld3 was added to an S-phase protein extract and tested for interaction with Cdc45. Immunoprecipitated proteins were analysed by immunoblot.

G    Reaction scheme for *in vitro* replication reactions. Reaction mixtures were removed after each step.

H    Reaction performed as in (G). Nascent DNA was separated in a 0.7% alkaline agarose gel in this and all subsequent replication reactions.

Mcm4 and 6 in a DDK-dependent manner (Fig 4B), and the Sld3-6E mutant was defective in binding these subunits (Fig 4D).

Dbf4-dependent kinase may promote Sld3 binding by inducing an allosteric change in Mcm subunits, revealing Sld3-binding sites. Alternatively, Sld3 may bind directly to DDK-dependent phosphopeptides. To begin to address this, we introduced a TEV cleavage site into the middle of Mcm6 (Fig EV4A) and tested for Sld3/7 binding to fragments of Mcm6, produced by cleavage with TEV protease. Figure EV4B shows Sld3/7 binds specifically only to the N-terminal half of Mcm6 in a DDK- and phosphorylation-dependent manner.

The majority of phosphorylation sites identified to date within Mcm6 are located in the serine-/threonine-rich extreme N-terminus of the protein (Randell *et al*, 2010). Figure EV4C shows that deletion of this domain (ΔN), or mutation of 25 potential DDK phosphorylation sites to alanine (25A) within this region, reduced but did not eliminate DDK-dependent Sld3/7 binding (Fig EV4C).

That the unstructured N-terminus of Mcm6 contributed to Sld3 binding suggested a direct interaction with phosphopeptides. However, because its deletion did not completely prevent binding

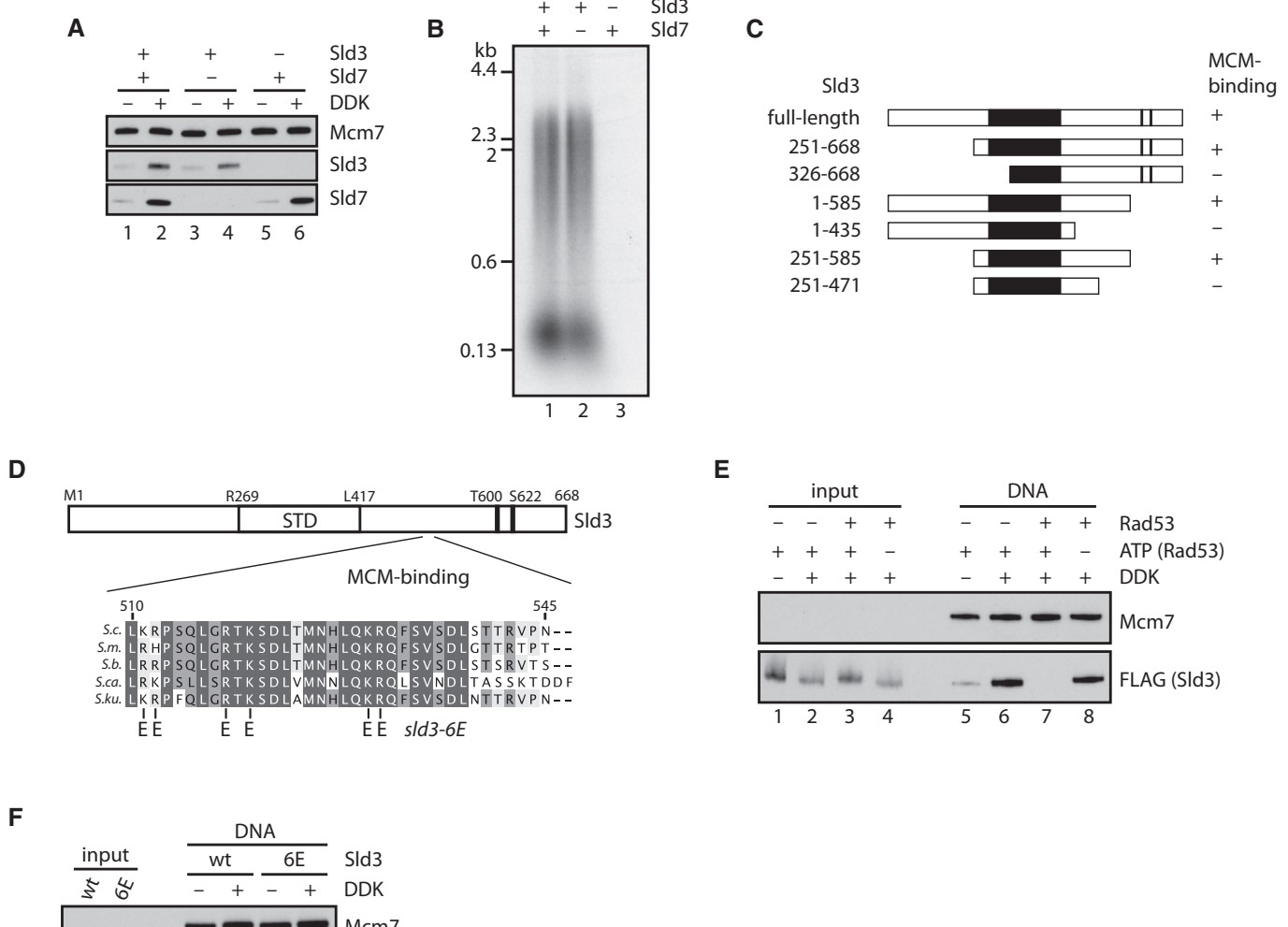

**Figure 2. Mapping the MCM-binding activity in Sld3/7.**

A Reaction performed as in Fig 1A, except Cdc45 was omitted.

B Reaction performed as in Fig 1G, with the indicated proteins omitted.

C Schematic depicting MCM binding by Sld3 fragments. Two essential CDK phosphorylation sites are shown by vertical bars. The Sld3–Treslin domain (S.T.D.) is shown as a black box. See also Fig EV1.

D Schematic showing the position of MCM-interacting region in Sld3. Alignments were generated as in Fig 1E. Residues were substituted for Glu as indicated.

E Reaction performed as in (A), except Sld3/7 was incubated with Rad53 before addition to reactions. ATP was omitted from this pre-incubation as indicated.

F Sld3 recruitment reaction performed as in (A). Sld7 was omitted from this experiment.

(Fig EV4C) additional phosphopeptides outside this domain must also exist. To identify these, we used a peptide array approach in which overlapping peptides attached to a membrane, either containing singly phosphorylated serine or threonine residues or unphosphorylated controls, were tested for binding to Sld3. These data, which help confirm that Sld3 does, indeed, bind directly to multiple phosphopeptides in Mcm6, are shown in detail in Fig EV5, with further description in Appendix Supplementary Text. We confirmed some of these interactions using HPLC-purified peptides attached to magnetic beads. Figure 4E shows an example of Sld3 binding to an Mcm6 peptide in a phosphorylation-dependent

manner. Mutation of 11 of the serine/threonine residues identified from this analysis to alanine, in combination with deletion of the N-terminal domain (ΔN+11A), completely prevented DDK-dependent Sld3/7 binding to Mcm6 (Fig 4F).

Given the phosphopeptide-binding activity of Sld3, and the apparent redundancy between binding sites, we next tested the Mcm4-25A mutant described previously in which 25 potential DDK sites were mutated to alanine (Randell *et al*, 2010) and found it too was defective in Sld3/7 binding (Fig 4F). Importantly, a number of the DDK sites mutated in Mcm4-25A interacted with Sld3 on a peptide array (Fig EV5G and H). Neither individual Mcm mutant

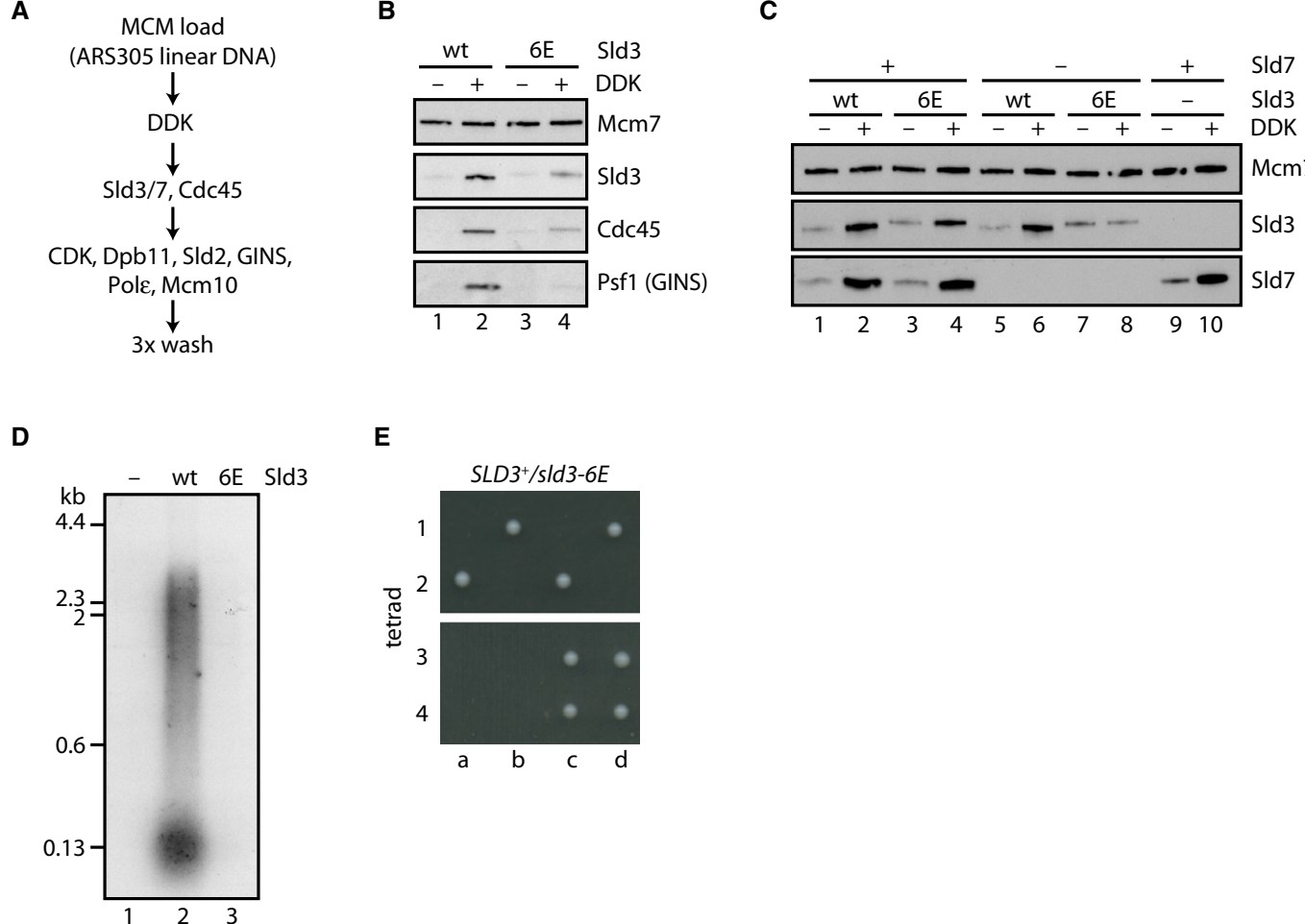

**Figure 3.   MCM binding is an essential function of Sld3.**

A   Reaction scheme for complete firing-factor recruitment reaction conducted on ARS305 linear DNA.
B   Recruitment reaction performed as described in (A).
C   Sld3/7 recruitment reaction conducted as in Fig 2A.
D   Replication reaction performed as in Fig 1G.
E   Representative tetrads from dissection of *SLD3+/sld3-6E* heterozygotes.

impacted on the ability of the other Mcm subunit to interact with Sld3/7 (Fig 4F), showing that Sld3/7 binds directly to both Mcm4 and Mcm6 in a DDK-dependent manner.

We next evaluated the effect of the Sld3-binding mutations in Mcm4/6 on DNA replication. The Mcm6-ΔN+11A mutant had little effect on Sld3/7 recruitment to the double hexamer, whilst the Mcm4-25A mutant reduced Sld3/7 recruitment (Fig 5A). The combination of the two mutants reduced Sld3/7 recruitment to near background levels (Fig 5A), indicating that Mcm4 and 6 are the key subunits for Sld3/7 binding to the double hexamer. Similarly, Fig 5B and C shows that the Mcm6-ΔN+11A mutant supported near wild-type levels of DNA replication, the Mcm4-25A mutant showed reduced replication and the double mutant supported even less replication. Consistent with this, the Mcm6-ΔN+11A mutant supported wild-type colony formation, the Mcm4-25A mutant exhibited a slow growth phenotype, and the double mutant showed synthetic lethality (Fig 5D). These results show a strong correlation

between Sld3/7 binding, DNA replication *in vitro* and survival *in vivo*, consistent with the idea that DDK-dependent Sld3/7 binding to MCM is critical for DNA replication. Furthermore, these data are consistent with Mcm4 being the most important DDK substrate for both Sld3/7 recruitment and replication initiation, in agreement with previously published data (Randell *et al*, 2010; Sheu & Stillman, 2010).

## MCM phosphorylation is not required downstream of initiation

The data presented thus far indicate that an essential function of DDK during DNA replication is to promote the interaction between Sld3 and Mcm4/6. Previous work has shown that Sld3 and DDK are each essential for helicase activation at origins, but are not required for continued DNA synthesis after initiation (Bousset & Diffley, 1998; Donaldson *et al*, 1998; Kanemaki & Labib, 2006). However, it was formally possible that MCM phosphorylation might fulfil some

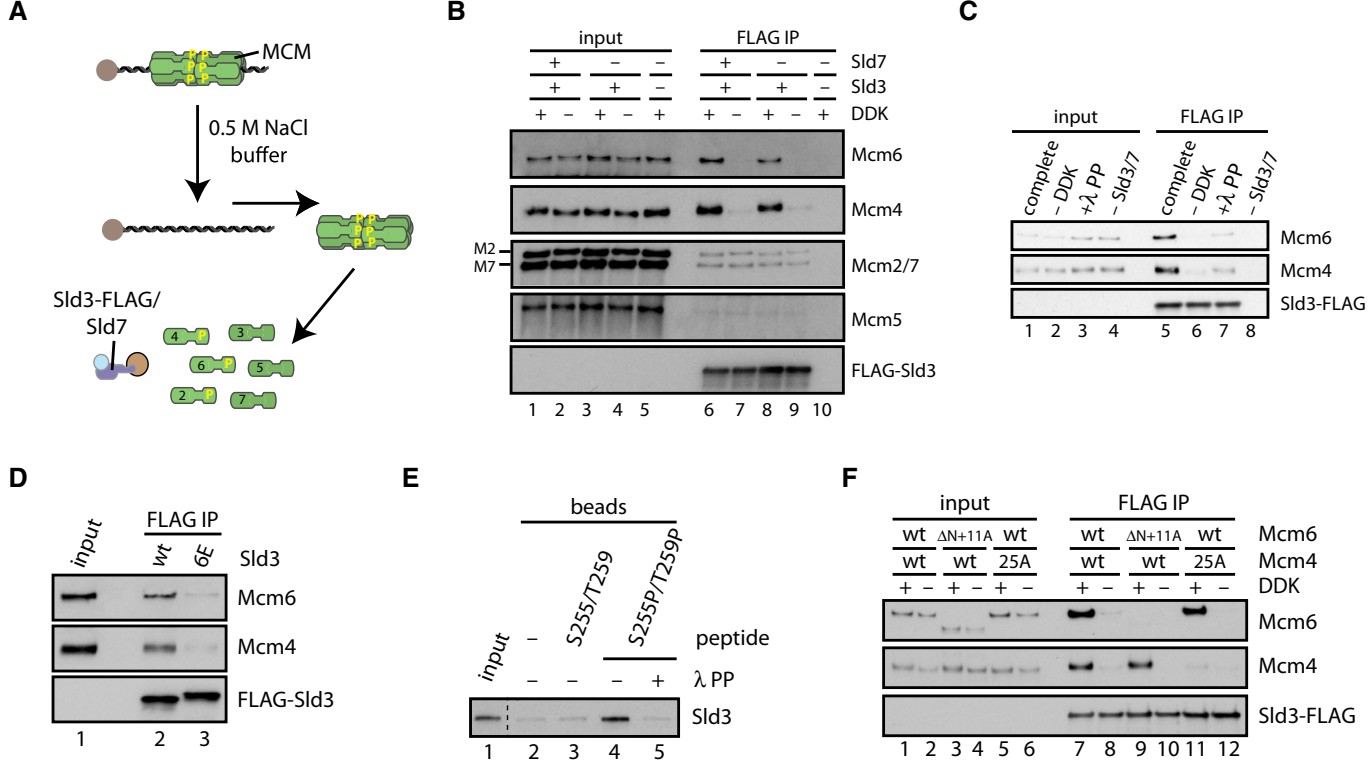

**Figure 4. Sld3 binds to phosphorylated Mcm4/Mcm6.**

A    Reaction scheme to examine interactions of Sld3/7 with individual Mcm subunits. Phosphorylated MCM bound to DNA was incubated in buffer containing 0.5 M NaCl to cause dissociation into individual subunits. Interactions with Sld3/7-coupled beads were then examined.

B–D    Immunoblot of reactions performed as described in (A). In (B), either reconstituted Sld3/7 complex (lanes 6 and 7) or Sld3 alone (lanes 8 and 9) was coupled to beads. In (C), MCM was treated with lambda protein phosphatase after dissociation from DNA (lanes 3 and 7). In (D), Sld7 was omitted.

E    30-residue phosphorylated Mcm6 peptides were bound to streptavidin-coated beads and used as bait to pull-down Sld3. Peptide-coupled beads were dephosphorylated with lambda protein phosphatase as indicated.

F    Reaction conducted as in (A), using wild-type (wt) or the indicated S/T-A substitution mutant versions of Mcm4 and Mcm6. Positions of mutations in Mcm6 are summarised in Fig EV5C.

additional function(s) downstream of initiation, and maintenance of MCM phosphorylation might thus be required for continued elongation by CMG.

To examine this, we asked whether dephosphorylation of MCM after initiation prevented subsequent elongation (Fig 6). We used the fact that replication on circular plasmids initiates efficiently without topoisomerase, but stalls within the first kilobase of synthesis because of the accumulation of positive supercoils (Yeeles *et al*, 2015). Subsequent addition of topoisomerase allows these stalled forks, which were pulse-labelled during the replication stalling, to resume replication (Fig 6C (ii) lanes 1,2). After DDK phosphorylation of loaded MCM, we treated samples with lambda protein phosphatase either before adding firing factors (Fig 6A (i)) or after stalling pulse-labelled forks by topoisomerase omission (Fig 6A (ii)). Phosphatase was then removed, and the remaining factors required for complete replication were added. In both cases, lambda protein phosphatase dephosphorylated Mcm4 and Mcm6, as evidenced by return to the faster migrating form in SDS–PAGE (Fig 6B). Phosphatase treatment before addition of firing factors blocked replication (Fig 6C (i) lanes 2,3), but phosphatase treatment after fork stalling had little or no effect on the resumption of

replication (Fig 6C (ii), compare lanes 2 and 4). Thus, like DDK and Sld3, MCM phosphorylation appears to be essential for initial helicase activation, but not for continued DNA synthesis after initiation in this system.

## Phosphomimicking MCM mutants support DDK-independent replication

If promoting the interaction between Sld3 and Mcm4/6 is the only essential function of DDK in replication, then bypassing the requirement for DDK in Sld3 recruitment should facilitate DDK-independent replication. To test this hypothesis, we generated "phosphomimicking" mutants in which serine and threonine residues in the N-termini of Mcm4 and 6 were changed to negatively charged aspartate residues. The various alanine-substitution mutants in Figs 4 and 5 all assembled into double hexamer with wild-type efficiency (Fig EV6A). However, although the aspartate substitution mutants formed stable MCM•Cdt1 complexes, they all showed defects in double-hexamer assembly: the individual Mcm6-25D and Mcm4-25D mutants showed partial defects, whilst the combined Mcm4-25D and Mcm6-25D mutants were almost completely

defective in loading (Fig EV6B). We generated a Mcm4-14D mutant (Randell *et al*, 2010) in which a subset of the Mcm4 DDK sites were mutated and combined this with the Mcm6-25D mutant. Although

this combination reduced MCM loading further than either single-25D mutant, there was still some loading (Fig EV6).

Figure 7A shows that Sld3/7 was recruited to each of the individual-25D mutants, and the combined-25D/14D mutants even in the absence of DDK (lanes 2–4). Levels of recruitment were above that seen with wild-type MCM without DDK (lane 1), but somewhat below the level seen with wild-type MCM after DDK phosphorylation (lane 5). This DDK-independent binding to MCM did not occur with the Sld3-6E mutant (Fig 7B), which also cannot bind DDK-phosphorylated MCM (Fig 2F). Each of the individual-25D and the combined-25D/14D mutants supported some level of CMG assembly in the absence of DDK (Fig 7C, lanes 2–4), in contrast to wild-type MCM (Fig 7C, lane 1).

The combined Mcm6-25D and Mcm4-14D mutant supported replication in the absence of DDK at approximately 60% the level achieved by the wild-type MCM complex with DDK (Fig 7D). Figure EV7A shows that the individual Mcm6-25D and Mcm4-25D mutants supported lesser, but detectable levels of replication without DDK. Despite supporting similar levels of Sld3/7 recruitment (Fig 7A), Mcm4-25D supported greater levels of replication than Mcm6-25D (Fig EV7A). We suggest that the Sld3–Mcm4 interaction is more efficient at promoting origin firing than the equivalent Sld3–Mcm6 interaction, which is also consistent with the phenotypes of the alanine-substitution mutants. Thus, similar to the situation with the Sld3-6E mutant, which can be recruited to MCM via Sld7, but cannot support replication (Fig 3), we suggest that "how" and "where" Sld3 is recruited to MCM is likely to be critical for efficient replication.

Previous work has shown that the phosphomimicking mutant of Mcm4 could suppress a *cdc7* mutant *in vivo* (Randell *et al*, 2010). Figure EV7B confirms this and shows that an Mcm6 phosphomimicking mutant can also partially suppress the temperature sensitivity of the *cdc7* mutant. Together, these results show that recruiting Sld3/7 to MCM with Mcm4/6 phosphomimicking mutants is

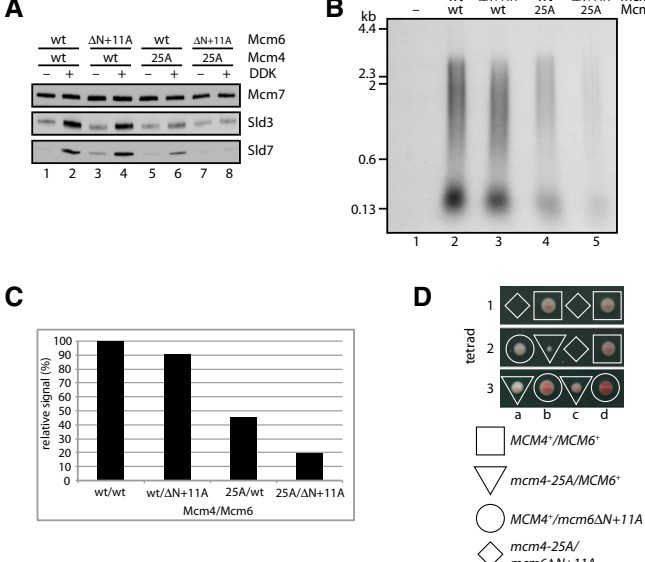

**Figure 5.  Sld3-binding mutants in Mcm4/Mcm6 are defective in replication.**

A   Sld3/7 recruitment reaction performed as in Fig 2A.

B   Replication reaction performed as in Fig 1G. Experiments in (A) and (B) used MCM containing wild-type (wt) or the indicated S/T-A substitution mutant versions of Mcm4 and Mcm6.

C   Quantification of (B) performed using ImageQuant software (GE Healthcare).

D   Representative tetrads from diploids heterozygous for *mcm4-25A* and *mcm6ΔN+11A*.

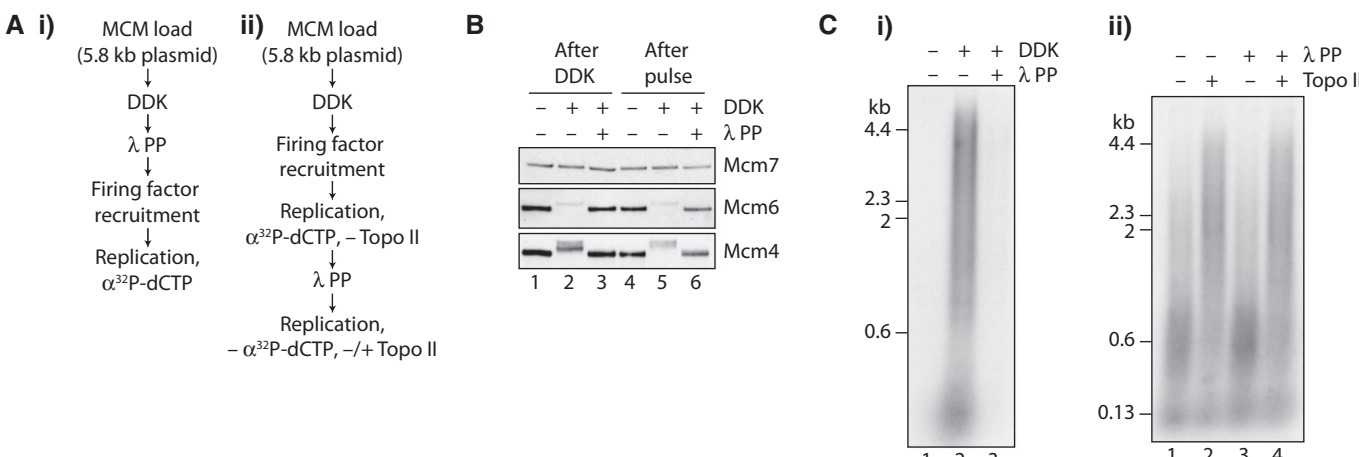

**Figure 6.   MCM phosphorylation is not required downstream of initiation.**

A   Reaction schemes for experiments in (B) and (C).

B   Immunoblots of reactions performed as in (A) part i (lanes 1–3) or ii (lanes 4–6).

C   Reactions in i and ii were performed as described in (A) parts i and ii, respectively.

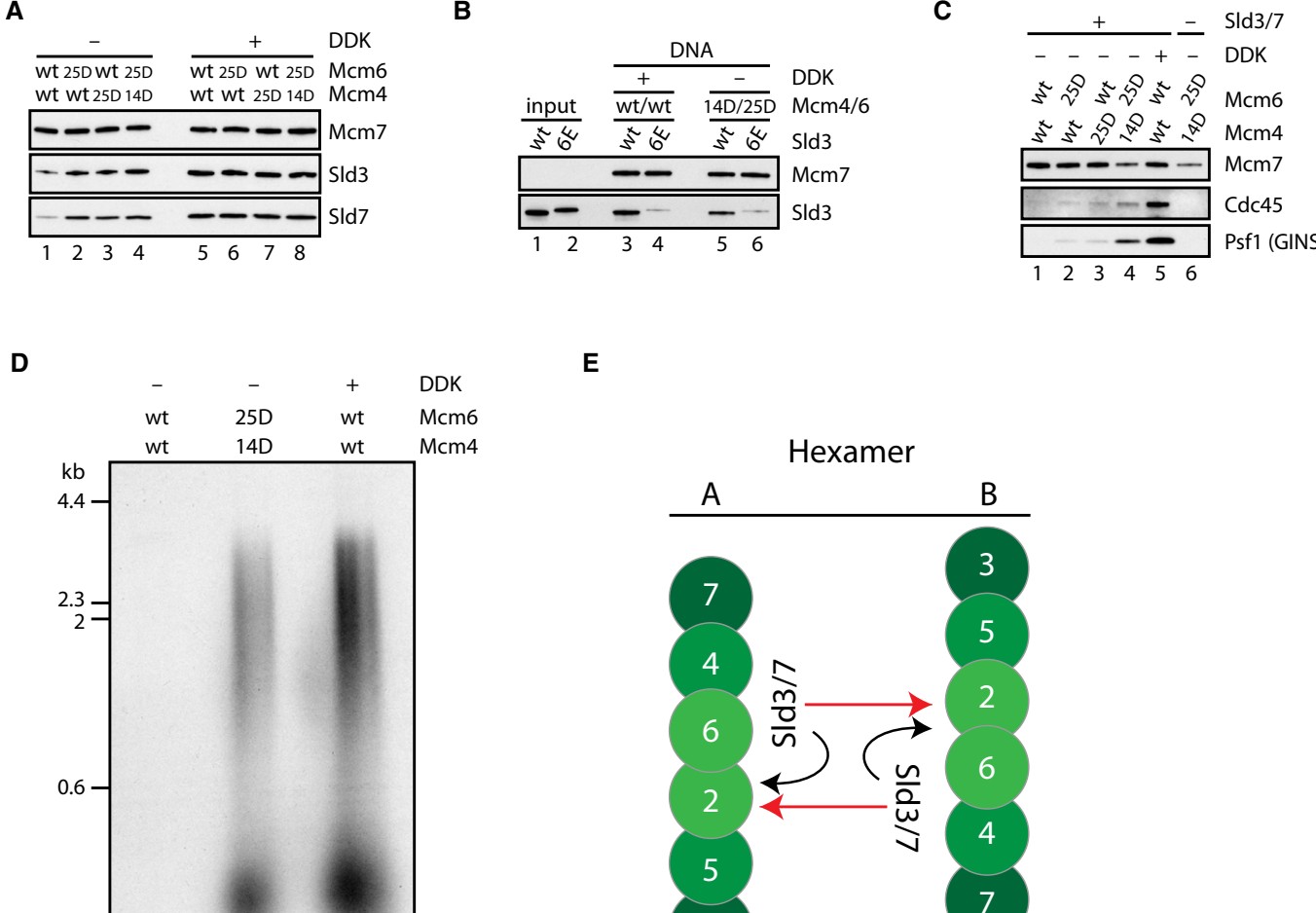

**Figure 7. DDK-independent replication.**

A, B  Recruitment reactions performed as in Fig 2A. Sld7 was omitted in (B).

C      Recruitment reaction conducted as in Fig 3A, with washes performed at 0.3 M KCl.

D      Replication reaction performed as in Fig 1G.

E      Model for Sld3/7-dependent Cdc45 recruitment to the MCM double hexamer. Sld3/7 may promote intra (black arrows)- or inter (red arrows)-hexamer Cdc45 recruitment. See text for full details.

Data information: Experiments in (A–D) used MCM containing wild-type (wt) or S/T-D substitution mutants of Mcm4 and Mcm6 as indicated. In (D), MCM loading was equalised across samples by using fivefold more mutant than wild-type MCM protein, as is shown in Fig EV6C.

sufficient to support DDK-independent replication, suggesting that Sld3 binding to Mcm4/6 is the main function of DDK in replication.

# Discussion

Our results show that Sld3 is a phosphopeptide-binding protein whose binding to DDK-phosphorylated Mcm4 or Mcm6 is required to recruit Cdc45 for the initiation of DNA replication. These results explain how DDK promotes the initiation of eukaryotic DNA replication at a molecular level. Given that Sld3 does not appear to be related to any known phosphopeptide-binding proteins, we suggest that Sld3 contains a novel phosphopeptide-binding domain.

The ability of Sld3 to bind the phosphomimicking mutants of Mcm4 and Mcm6 suggests the negative charge of the phospho-amino acid is critical for binding. Nonetheless, Sld3 does not bind to all phosphopeptides (Fig EV5) suggesting local sequence context provides binding specificity. Further work is required to identify specificity determinants in primary amino acid sequences. Sld3 binds better to multiply phosphorylated peptides (Fig EV5I), and binding to phosphopeptides involves at least six lysine residues in Sld3 (Figs 2 and EV2) suggesting that the phosphopeptide-binding site of Sld3 may have evolved to preferentially recognise peptides containing multiple phospho-amino acids, consistent with the fact that Mcm4 and Mcm6 are phosphorylated on many sites by DDK. It will be interesting to determine whether the human Sld3

orthologue Treslin/TICRR contains an orthologous domain and is also a reader of DDK phosphorylation. In contrast to recently published work, our results show that Sld3 is not recruited to MCM via DDK-independent interaction with Mcm2 (Herrera *et al*, 2015).

It is likely that we do not know all DDK sites in MCM subunits due to incomplete coverage in mass spectrometry (Randell *et al*, 2010). In addition, it is likely we have not identified all possible Sld3-binding phosphopeptides since our peptide arrays contained only singly phosphorylated peptides. Consequently, we do not, at present, know whether all DDK phosphorylation sites generate Sld3-binding sites or whether all Sld3-binding sites contain DDK sites. Moreover, given the inherent complexity in the functional redundancy, we do not know if all DDK/Sld3 sites are functionally equivalent. Further work will yield a fuller understanding of these issues.

Our results show that Sld3 is a critical integrator of signals from the CDK, DDK and DNA damage checkpoint protein kinase pathways into the binary decision to fire an origin. We suggest that the preference of Sld3 for multiple phosphorylated peptides together with the fact that both Dbf4 and Sld3 are limiting for DNA replication in S phase (Mantiero *et al*, 2011; Tanaka *et al*, 2011a) contribute to converting analogue signals from multiple protein kinase pathways into an on/off switch for initiation at individual origins, perhaps analogous to the way in which multiple CDK phosphorylations contribute to the switch-like degradation of the Sic1 CDK inhibitor (Nash *et al*, 2001). This may also contribute to converting small differences in affinities for firing factors into a robust temporal programme of origin firing (Douglas & Diffley, 2012).

Ultimately, firing factors promote CMG formation, in which Cdc45 and GINS bind stably to Mcm2 and Mcm5, respectively, bridging the weak Mcm2–Mcm5 interface (Costa *et al*, 2011). By binding Mcm4 and Mcm6, Sld3 may promote Cdc45 binding to Mcm2 within a single hexamer (Fig 7E, black arrows). However, given that DDK phosphorylation of Mcm4 appears more important than Mcm6 for Sld3/7 recruitment, but Mcm4 is not adjacent to Mcm2 in a single hexamer, we suggest that Sld3 promotes Cdc45 recruitment to Mcm2 on the opposing hexamer (Fig 7E, red arrows), which resides opposite Mcm4 and Mcm6 across the double-hexamer junction (Costa *et al*, 2014; Sun *et al*, 2014). This may help ensure Cdc45 is only recruited to loaded MCM double hexamer. Related to this, recent evidence indicates that two Sld3 molecules are connected by an Sld7 dimer within a single Sld3/7 complex (Itou *et al*, 2015); by linking two Sld3 molecules bound to phosphorylated peptides on opposing MCM hexamer, Sld7 may function to co-ordinate bi-directional CMG assembly and origin firing.

The DDK bypass mutants, especially the full phosphomimicking Mcm6-25D and Mcm4-25D combination, have defects in double-hexamer assembly. It is interesting to consider that the inability of DDK-phosphorylated Mcm4 and Mcm6 to load into double hexamer may play some minor, perhaps local, role in preventing re-initiation of DNA replication. In addition to the phosphomimicking DDK bypass mutants described here and in Randell *et al* (2010), mutants in Mcm4 and Mcm5 have been described which bypass the requirement for DDK (Hardy *et al*, 1997; Sheu & Stillman, 2010). These mutants may work by binding Sld3/7 even in the absence of DDK or may recruit Cdc45 in the absence of Sld3/7. Further work is required to understand the mechanism of bypass by these mutants.

## Materials and Methods

### Yeast strains and methods

Yeast strains were constructed and manipulated by standard genetic techniques. Details of all strains and vectors are given in Appendix Tables S1 and S2. Mutagenesis of *SLD3* at the endogenous locus was done by integration of a cassette derived from pFA6a-natNT2 (Janke *et al*, 2004). Full-length mutagenised *SLD3* was cloned upstream of *Nat-NT2*. The resultant *SLD3-Nat-NT2* cassette was amplified by PCR and then used to replace an endogenous copy of *SLD3* by transformation into a yeast diploid.

Mutagenesis of *MCM4* at the endogenous locus was done by integration of a cassette derived from pFA6a-kanMX6 (Longtine *et al*, 1998). Full-length mutagenised *MCM4* (including 500 bp of sequence found directly upstream of *MCM4* at the endogenous locus) was cloned upstream of *KanMX6*. The resultant *MCM4-KanMX6* cassette was then used to replace an endogenous copy of *MCM4* by transformation into a haploid yeast strain. Mutagenesis of *MCM6* at the endogenous locus was performed in the same way, except the targeting construct was derived from pFA6a-natNT2.

Endogenous *MCM4* and *MCM6* were C-terminally tagged with 3xFLAG by transformation with PCR products generated using pBP83 (Frigola *et al*, 2013) or pBP80-ADE2 as a template.

### Protein expression and purification

ORC, Cdc6, MCM•Cdt1, DDK, Rad53, Sld3/7, Cdc45, S-CDK, Dpb11, Sld2, GINS, Polε, Mcm10, Ctf4, TopoII, RPA and Polα/Primase were expressed and purified as previously described (Gilbert *et al*, 2001; Coster *et al*, 2014; On *et al*, 2014; Yeeles *et al*, 2015). Purification of MCM•Cdt1 complexes containing mutant versions of Mcm4/6 was performed as previously described for wild-type MCM•Cdt1 (Coster *et al*, 2014), except that samples were subjected to FLAG immunoprecipitation to deplete endogenous Mcm4-3xFLAG and/or Mcm6-3xFLAG prior to gel filtration. Peak fractions from calmodulin affinity purification were incubated with 0.3 ml packed bead volume of anti-FLAG M2 agarose (Sigma). After 45 min rotation at 4°C, the flow-through was collected, concentrated and subjected to gel filtration as described previously.

### Expression and purification of Sld3 or Sld7 from *E. coli*

BL21 CodonPlus RIL cells (Stratagene) were transformed with the relevant pGEX-6p-1/Sld3 or pGEX-6p-1/Sld7 plasmid (see Appendix Table S2). Transformant colonies were used to inoculate a 50 ml LB/ampicillin (50 µg/ml) culture, which was grown overnight at 37°C with shaking at 250 rpm. The following morning, the culture was diluted tenfold in 250 ml of LB/ampicillin (50 µg/ml)/chloramphenicol (35 µg/ml). The culture was left to grow at 25°C for ~4 h until an $OD_{600}$ of 0.7 was reached. 1 mM IPTG was added for 4 h at 25°C to induce expression. Cells were harvested by centrifugation at 2,070 *g* for 15 min in an SLA-3000 rotor (Thermo Scientific), and the resultant pellet was then washed once in ice-cold PBS.

For lysis, cell pellets were resuspended in 10 ml of buffer containing 25 mM HEPES-KOH pH 7.6, 0.05% (v/v) NP-40, 10% (v/v) glycerol, 500 mM KCl, 1 mM DTT (buffer A/500 mM KCl)

plus protease inhibitors (1 mM EDTA, 5 mM benzamidine–HCl, 1.5 μM pepstatin A, 0.5 mM AEBSF, 0.3 μM aprotinin, 3 mM PMSF, 2 μM leupeptin). Lysozyme was added to a final concentration of 500 μg/ml. The mixture was then left on ice for 20 min and subsequently sonicated for 2 min (25 s on, 5 s off) at setting 5 on a Sonicator Ultrasonic Processor XL (Misonix). Insoluble material was removed by centrifugation at 26,892 g for 15 min in an SS-34 rotor (Sorvall).

The supernatant was subjected to GST affinity purification by incubation with 0.8 ml packed bead volume of glutathione sepharose resin (GE Healthcare) that had been pre-washed in buffer A/500 mM KCl. The sample was rotated at 4°C for 2 h. Glutathione beads were recovered in a disposable gravity flow column and washed with 15 column volumes buffer A/500 mM KCl plus protease inhibitors followed by 5 column volumes buffer A/500 mM KCl without protease inhibitors. A 50% slurry was prepared using buffer A/500 mM KCl, and 50 μl preScission protease (GE Healthcare) was added. The mixture was rotated at 4°C overnight.

The following morning, the flow-through was collected and one further column volume of buffer A/500 mM KCl was passed over the column to elute any remaining cleaved protein. Peak fractions containing Sld3 or Sld7 were pooled, concentrated using an Amicon Ultra 30,000 MWCO centrifugal filter (Millipore) and stored in aliquots at −80°C. Purification of Sld3 fragments or mutants was performed as described above.

### DNA templates

One kb biotinylated linear ARS305 and 3.2 kb or 5.8 kb randomly biotinylated ARS1 circular templates were generated as previously described (Coster et al, 2014; Yeeles et al, 2015). Biotinylated DNA was coupled to streptavidin-coated M-280 dynabeads (Invitrogen) as previously described (Coster et al, 2014).

### Protein recruitment and replication reactions

All MCM loading and firing-factor recruitment reactions were performed using the ARS305 linear DNA template. MCM loading reactions were performed as described previously (Coster et al, 2014) with one modification. Proteins bound to DNA were eluted from beads by incubation in 10 μl buffer containing 45 mM HEPES-KOH pH 7.6, 300 mM KOAc, 5 mM Mg(OAc)$_2$, 10% (v/v) glycerol, 0.02% (v/v) NP-40, 5 mM CaCl$_2$ (wash buffer/300 mM KOAc/CaCl$_2$) and 1,000 units micrococcal nuclease (Sigma). Samples were incubated at 37°C for 2 min, beads were re-isolated on a magnetic rack, and the supernatant was then removed.

Standard firing-factor recruitment reactions were performed as previously described (Yeeles et al, 2015). Sld3/7 and Cdc45 recruitment reactions were performed as described for complete firing-factor recruitment reactions, except the "CDK step" was omitted and poly(dI/dC) was included at 150 ng/μl in the Sld3/7/Cdc45 step. Standard washes were performed in wash buffer/0.3 M K-glutamate unless stated in the figure legend.

For Fig 2E, 1 pmol Sld3/7 was pre-incubated with 0.1 pmol of Rad53 for 30 min at 25°C in a buffer containing 25 mM HEPES-KOH pH 7.6, 65 mM KOAc, 50 mM KCl, 10 mM magnesium acetate, 0.02% NP-40, 5% glycerol, 5 mM MgCl$_2$, 1 mM DTT and 5 mM

ATP. After 30 min, the concentration of KOAc was adjusted to 500 mM, and the reaction mixture was then added directly to DDK-phosphorylated MCM. Final washes were performed at 500 mM KOAc.

Replication reactions were performed using 3.2 kb or 5.8 kb randomly biotinylated ARS1 circular templates as described previously (Yeeles et al, 2015).

For Fig 6B and C beads were isolated after DDK treatment and subjected to 3 × 150 μl washes in wash buffer/600 mM NaCl, followed by a single 150 μl wash in wash buffer/150 mM K-glutamate. Firing-factor recruitment was then conducted in a single 10 min step at 25°C in a buffer (20 μl) containing 40 mM HEPES–KOH pH 7.6, 250 mM K-glutamate, 10 mM Mg(OAc)$_2$, 0.02% NP-40, 8% glycerol, 400 μg/ml BSA (NEB), 2 mM DTT, 5 mM ATP, 26 nM Sld3/7, 50 nM Cdc45, 40 nM Dpb11, 62 nM Sld2, 30 nM Polε, 210 nM GINS, 5 nM Mcm10 and 30 nM S-CDK. For pulse-chase reactions, the supernatant was removed and replaced with a standard replication reaction mix where the dCTP concentration was reduced to 4 μM and from which TopoII was omitted. Following a 20-min incubation at 30°C, beads were isolated, washed once with 150 μl wash buffer/150 mM K-glutamate and resuspended in 20 μl 40 mM HEPES–KOH pH 7.6, 150 mM K-glutamate, 10 mM Mg(OAc)$_2$, 5% glycerol, 2 mM DTT, 1 mM manganese chloride, 50 μM ATP, 4 μM dATP, dTTP, dCTP, dGTP and 1.2 μg lambda protein phosphatase. Reactions were incubated at 30°C for 10 min before beads were isolated, washed twice in 150 μl wash buffer/150 mM K-glutamate and resuspended in standard replication reaction buffer (20 μl) containing 5 nM Mcm10, 50 nM RPA, 20 nM Polα, 30 nM Ctf4 and 25 nM TopoII but lacking labelled dCTP. Reactions were incubated for 45 min at 30°C before being terminated and processed as described for standard replication reactions. When DDK-phosphorylated MCM was incubated with lambda protein phosphatase prior to firing-factor recruitment, beads were isolated following the DDK treatment and subsequent washes and were resuspended in buffer containing lambda protein phosphatase as described for pulse-chase reactions. Following 10-min incubation at 30°C, the beads were isolated, washed twice in 150 μl wash buffer / 150 mM K-glutamate before firing-factor recruitment was conducted in a single step as described for pulse-chase reactions. The replication step was then performed as described for standard replication reactions for 45 min at 30°C.

For recruitment and replication experiments using individually purified Sld3 and Sld7, the Sld3/7 complex was reconstituted by incubating Sld3 and Sld7 together at 4°C for 30 min prior to addition to reactions.

### Interaction of Sld3/7 with Cdc45 in yeast whole-cell extracts

S-phase extract, prepared as previously described from yKO3 (On et al, 2014), was initially depleted of Sld3-13Myc by incubation with an equal volume of anti-Myc magnetic bead slurry (Origene) as extract. 2 × 1 h depletions were performed on ice with fresh anti-Myc beads added each time. 20 μl reactions were then assembled containing 40 mM HEPES-KOH pH 7.6, 8 mM MgCl$_2$, 40 mM creatine phosphate, 10 μg creatine phosphokinase, 1 mM DTT, 5 mM ATP and 250 μg yKO3 S-phase extract. A buffer containing 50 mM HEPES-KOH pH 7.6, 0.3 M K-glutamate, 5 mM

Mg(OAc)$_2$, 1 mM EDTA, 1 mM EGTA, 10% (v/v) glycerol and 3 mM DTT (buffer B) was added to adjust the final concentration of K-glutamate to 150 mM. 0.5 pmol wild-type or mutant FLAG-Sld3 and 0.5 pmol Sld7 were then added and the reaction incubated on ice for 15 min. The reaction mixture was then added to 2.5 μl magnetic anti-FLAG beads (Sigma), which had been pre-washed in buffer B. Samples were incubated for 1 h on ice with occasional agitation. Beads were then isolated on a magnetic rack, the supernatant removed, and each sample was subjected to 2 × 200 μl washes in wash buffer/150 mM K-glutamate. The beads were then resuspended in Laemmli sample buffer and boiled for subsequent analysis by SDS–PAGE.

### Interaction of Sld3/7 with individual Mcm subunits

A 120 μl MCM loading reaction (using 6 μl of linear ARS305 DNA beads) was prepared per sample as previously described. After the MCM loading step, each sample was subjected to sequential 400 μl wash steps using wash buffer/300 mM KOAc, wash buffer/500 mM NaCl and wash buffer/100 mM KOAc (in that order). DDK phosphorylation (120 μl reaction volume) was then performed as previously described. Subsequently, beads were re-isolated, and each sample was washed in 400 μl wash buffer/500 mM NaCl. The wash buffer was removed, and each beads sample was resuspended in 65 μl wash buffer/500 mM NaCl and incubated on ice overnight. In experiments using TEV-cleavable Mcm6, 0.5 nmol HIS$_6$-TEV was added at this stage of the reaction.

The following morning, the beads were isolated on a magnetic rack and a 60 μl supernatant fraction was recovered from each sample. A 10 μl input fraction was removed, and 75 μl saltless wash buffer was added to the remaining 50 μl sample to adjust the concentration of NaCl to 200 mM. The resultant 125 μl sample was then added to 2.5 μl magnetic anti-FLAG beads (Sigma), which had been pre-coupled to Sld3-3xFLAG/Sld7 or Sld3-FLAG. Samples were incubated for 1 h at 8°C with agitation at 1,100 rpm. Beads were then isolated on a magnetic rack, and each sample was subjected to 2 × 200 μl washes in wash buffer/200 mM NaCl. Input and beads samples were routinely dephosphorylated with lambda protein phosphatase (1 h, 30°C) prior to the addition of Laemmli sample buffer to prevent the masking of epitopes by phosphorylation during immunoblotting.

### Interaction of Sld3 with Mcm4/6 peptides

N-terminally biotinylated peptides and peptide arrays were synthesised by the Peptide Chemistry Laboratory at the Francis Crick Institute. For Mcm6 peptide arrays, a membrane was spotted with 18-residue peptides covering Mcm6 residues 1–486. For Mcm4 peptide arrays, a membrane was spotted with 18-residue peptides covering Mcm4 residues 1–183. A two-residue start increment was used between peptides, with singly phosphorylated and unphosphorylated versions of each peptide spotted next to one another. Peptides bound to membranes were solubilised by washing in 50% EtOH/10% acetic acid for 1 h. The membranes were washed briefly in TBST, then incubated at 4°C overnight in TBST/3% BSA. The following morning, membranes were washed briefly in a buffer containing 25 mM HEPES-KOH pH 7.6, 10 mM Mg(OAc)$_2$, 0.02% (v/v) NP-40, 5% (v/v) glycerol, 500 mM KOAc, 50 mM

KCl and 1 mM DTT (peptide array buffer) and then incubated for 1 h at room temperature with 40 ml peptide array buffer/3% BSA containing 2 nM FLAG-Sld3. Membranes were subsequently washed for 3 × 3 min in peptide array buffer, then probed with anti-FLAG antibody (Sigma) for 3 h at room temperature. Three consecutive 10-min washes in peptide array buffer were finally performed followed by application of the ECL chemiluminescence reagent (Pierce).

N-terminally biotinylated peptides were coupled to streptavidin-coated M-280 dynabeads (Invitrogen) by incubation of 40 pmol of peptide with 4 μl beads for 30 min at 20°C with agitation at 1,250 rpm. Peptide-coupled beads were subsequently washed in peptide array buffer. An 80 μl reaction mixture containing peptide array buffer and 5 nM FLAG-Sld3 was added to 4 μl beads, and the reaction incubated at 25°C for 10 min with agitation at 1,000 rpm. Beads were isolated on a magnetic rack and then subjected to 2 × 200 μl washes in peptide array buffer. The beads were then resuspended in Laemmli sample buffer and boiled for subsequent analysis by SDS–PAGE.

### Antibodies

Anti-Mcm2 (yN-19, sc-6680, Santa Cruz), anti-Mcm4 (yC-19, sc-6685 Santa Cruz), anti-Mcm5 (yN-19, sc-6686, Santa Cruz), anti-Mcm7 (yN-19, sc-6688, Santa Cruz), anti-Orc6 (SB49), anti-Cdc6 (9H8/5). FLAG-tagged proteins were detected with anti-FLAG M2 (Sigma). Mcm3 was detected with anti-CBP antibody (07-482, Merck Millipore). Polyclonal antibodies against Mcm6, Sld3, Sld7 and Cdc45 were described previously (On *et al*, 2014). Psf1 antibodies were a gift from K. Labib.

**Expanded View** for this article is available online.

## Acknowledgements

We are grateful to Anne Early for help with yeast strain construction, Max Douglas for helpful discussion, the Peptide Chemistry Laboratory at the Francis Crick Institute for peptide synthesis, Ali Alidoust and Namita Patel for growing yeast cultures, Philip Zegerman for Sld7 antibodies and Karim Labib for Psf1 antibodies. This work was supported by Cancer Research UK, the Francis Crick Institute, a FEBS Return-to-Europe fellowship to J.T.P.Y., an ERC grant (249883 – EUKDNAREP) and a Wellcome Senior Investigator Award (106252/Z/14/Z) to J.F.X.D.

## Author contributions

TDD and JFXD designed the study and prepared the manuscript. TDD and JTPY performed the experiments.

## Conflict of interest

The authors declare that they have no conflict of interest.

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
