## [Review Process File · The EMBO Journal]

Manuscript EMBO-2015-93552

Phosphopeptide binding by Sld3 links Dbf4-dependent kinase to MCM replicative helicase activation

Tom D. Deegan, Joseph T. P. Yeeles and John F. X. Diffley

Corresponding author: John F. X. Diffley, The Francis Crick Institute

Review timeline:

Submission date:	23 November 2015
Editorial Decision:	16 December 2015
Revision received:	07 January 2016
Editorial Decision:	14 January 2016
Revision received:	02 February 2016
Accepted:	03 February 2016

*Editor: Hartmut Vodermaier***Transaction Report:**

1st Editorial Decision

16 December 2015

Thank you again for submitting your manuscript on Sld3 phosphorylation recognition for our consideration. We have now received the comments of three expert referees, and I am pleased to inform you that all of them consider your analyses well-conducted and the results of interest and importance. Pending adequate revision of number of specific points raised by all three reviewers, mostly concerning aspects of interpretation/discussion and manuscript presentation/organization, we shall therefore be happy to consider the study further for publication in The EMBO Journal.

Should you have any additional question regarding this decision or the referee reports, please do not hesitate to contact me. I would like to thank you once more for the opportunity to consider this work for The EMBO Journal, and look forward to your revision.

REFeree COMMENTS

Referee #1:

The initiation step of DNA replication in eukaryotic cells requires two protein-kinases, DDK and CDK. In yeast, CDK phosphorylates Sld2 and Sld3 for binding to Dpb11 to initiate DNA replication. Because DDK phosphorylates Mcm2, Mcm4 and Mcm6, components of the pre-RC formed on replication origins, and promotes the association between Sld3 and Cdc45, and origins, it has been thought that phosphorylation of Mcms recruits Sld3 and Cdc45. This paper revealed that Sld3 binds DDK-phosphorylated Mcm4 and Mcm6 using reconstituted *in vitro* assays. Sld3 binds to Cdc45 as well as Mcm2-7. The authors first separated these bindings by constructing mutant Sld3. The mutations outside of the known Cdc45-binding domain reduced the affinity of Sld3 to Mcm2-7.

They determined the subunits of Mcm2-7 bound by Sld3 by a sophisticated way and further determined the specific phosphorylation sites of Mcm4 and Mcm6 for binding to Sld3. Finally, they showed that the phospho-mimicking mutants of Sld3 partially bypass the requirements of DDK for Sld3 association with origin DNA and DNA replication *in vitro*. This is confirmed *in vivo*. Then, the authors argue that Sld3 is a protein binding to DDK-phosphorylated peptides and thus a key protein sensing DDK signals. This is an important finding not only for the mechanism of DNA replication but also regulation of the cell cycle. Besides the results described above, two more important findings for DNA replication and checkpoint control are reported; the interaction between Sld3 and Mcm2-7 is inhibited by checkpoint kinase, Rad53, and the elongation step of DNA replication require no more DDK-phosphorylation. The quality of experiments is high; each experiment is well designed to draw the results and probably only few labs are able to perform this kind high quality experiments because of the requirements of many purified proteins and accumulated skills for the *in vitro* assays. Thus this paper is worth publishing in the EMBO Journal. However, there are some points should be discussed before publication.

(Major points)

1. Is it really phosphorylation specific? The authors used phosphorylated peptides and mutated Sld3 proteins for this analysis. However, it is still conceivable that the surface charge (acidophilic charges caused by phosphorylation) of the specific sequence in Mcms recruits Sld3. Although this is a difficult question, the authors may discuss the difference of affinities between phospho-mimicking mutant and phosphorylated wild type proteins.
2. There are two discrepancies between the results of this paper and previous *in vivo* observations. First, *in vivo* analysis of budding yeast showed that Sld3 associates with origins in a Cdc45-dependent manner (Kamimura et al., 2001) while the authors showed in this paper that Sld3 alone can bind to Mcm2-7 *in vitro*. Second, Sld7, which binds to Sld3, was reported to associate with origins in an Sld3-dependent manner (Tanaka et al., 2011). Although I do trust the results obtained in this paper (that is, Sld3 is a key protein recognizing the phosphorylated Mcms), these discrepancies can be explained if we assume some other factors *in vivo*, for example. Thus, the authors should discuss this possibility carefully.

(Minor and miscellaneous points)

1. The authors discuss the interaction between Sld3 and Mcm2-7 in Fig.7. The published partial structure of Sld3/Sld7 and the proposal model described in the same report (Itou et al., *Acta Cryst. D* 71, 1649-1656, 2015) may further support the authors' model.
2. p.5 line 4, "to be" is repeated.
3. p.6, 2nd paragraph, line 3, "lanes 6 and 8" should read "lanes 4 and 8".
4. Some references lack pages, please fix them.
5. There are many supplementary data. If the page limit allows, I recommend showing some of them in main results (for example, peptide screenings). It is much easier for audience to follow the text.

Referee #2:

The manuscript by Deegan et al described the roles of Sld3 in assembly of the replication initiation complex using *in vitro* DNA replication system of budding yeast. Sld3 is an essential initiation factor for DNA replication and loaded onto the replication origins in a manner dependent on Dbf4-dependent kinase (DDK). However, the mechanism how Sld3 is loaded has not been shown. The authors identified the regions of Sld3 participated in the interactions with Cdc45 and Mcm2-7 complex, respectively. They demonstrated that Sld3 is a phosphopeptide-binding protein and it interacted specifically with phosphorylated Mcm4 and Mcm6 in a Mcm2-7 double hexamer. Sld3 interacted with specific phosphorylated-polypeptides from Mcm4 and Mcm6. Alanine substitutions for the phosphorylation sites of Mcm4 and Mcm6 impaired Sld3 loading and the subsequent recruitment of Cdc45 and GINS, resulting in failure of *in vitro* DNA replication and loss of the viability. Consistently, aspartic acid substitutions for the corresponding residues of Mcm4 and Mcm6 partially bypassed the requirement of DDK both *in vitro* and *in vivo*, suggesting that the essential role of DDK is to phosphorylate Mcm4 and Mcm6 for the interactions with Sld3. Although this work contains confirmative results of the previous finding by Randell et al (2010), it provided the novel important understanding in regulation of DNA replication and genome maintenance.

Major comments:

There are two major concerns.

The most important finding of this paper is that Sld3 binds specifically to DDK-phosphorylated polypeptides of Mcm4 and Mcm6. Nonetheless, contributions of DDK-phosphorylation to Sld3 binding were not clearly distinguished from DDK-independent phosphorylation at polypeptide level. In addition, the detailed results of the phosphorylated peptides of Mcm4, which seems to be more important than Mcm6, were not presented. The second problem is that the manuscript is premature in logical writing. The aim of the experiment was not properly presented at several places and the important results were not precisely described. Extensive rewriting is required. The individual comments are shown below.

1. As stated in the Abstract, the authors found that Sld3 binds specifically to DDK-phosphorylated peptides from Mcm4 and Mcm6. However, this conclusion is not readily supported by the results presented in the main figures or in the supporting results (Fig EV5). I felt difficulty in distinguishing the phosphorylated residues that interacted with Sld3 (Fig EV5A and B). For example, T66P and S69S exhibit similar intensity in EV5A but only T66 is listed in EV5B. In addition, it is not clear how DDK-phosphorylation contributes to the interaction of specific polypeptides with Sld3. The complete set of the results and the criterion for the positive interaction should be included. More importantly, the detailed results of Mcm4 polypeptides were not presented, despite of importance of Mcm4 in DDK-dependent regulation. Comprehensive presentation of the DDK-dependent and DDK-independent phosphorylation sites found in the Sld3-interacting polypeptides might be required.
2. Relating with the above points, some of the supporting data (Fig EV5) are better to be included in the main figures. The authors need to provide enough information for individual phosphorylation sites.
3. I recommend to include Fig EV3 B and C instead of Fig 4B, because the latter figures demonstrated the preferential interactions of Mcm4 and Mcm6 with Sld3.
4. I think the manuscripts need to be reorganized to clarify the logical flow. For example, the second section "MCM binding is an essential function of Sld3", was difficult to follow. In page 5 line 20, the aim of the experiments should be stated before describing the results. Then, the experiments for Rad53 (Fig 2E) from page 5 line 26 to page 6 line 6 should be described later, for example, after the results of amino acid substitutions (Fig 2F and Fig 3). Furthermore, the aim of the experiment is again missing at the beginning of the third paragraph (page 6 line 13). Despite of importance of Fig 3D and 3E, the data were not precisely described. This section needs to be reorganized.
5. Page 8 line 3, the aim of the experiment (Fig 5) should be stated before describing the results.
6. Page 8 lines 10-14, the authors need to mention the significance of severer effects of the Mcm4 mutant than the Mcm6 mutant.
7. Fig. 6 might be moved into the supporting data, because the question raised seems to be less interesting than others.
8. Page 10 lines 10-14, the results in Fig. 7C and 7D should be described separately and more precisely. The individual mcm6-25D and mcm4-25D mutants promoted Sld3 recruitment at a similar level with the double mutant in the absence of DDK (Fig 7A), whereas they did not efficiently support Cdc45 or GINS loading compared with the double mutant (Fig 7C). Do these results suggest some other roles of DDK-phosphorylation on Mcm4 and Mcm6? The authors need to explain the difference. In addition, the description "at levels that approached that achieved by the wild type MCM complex with DDK" did not precisely reflect the data and should be improved.
9. Because Fig 7D showed only partial information, it should be replaced with Fig EV7A.
10. Page 10 lines 18-19 and Fig EV7B, suppression of cdc7-4 defect by mcm6-18D at 30{degree sign}C was not convincing. Does mcm6-18D in combination with mcm4-25D show better

suppression of *cdc7-4* than *mcm4-25D* alone? Since some data are confirmation of the previous work, the clear evidence for contribution of Mcm6-phosphorylation should be provided for the novelty.

Minor comments

1. Page 5 lines 5-6, "Sld3 origin association clearly precedes Cdc45 (Nakajima et al., 2002) in fission yeast"; the report by Yabuuchi et al (2006) should be the appropriate reference, because it showed that Sld3 but not Cdc45 is loaded onto origin in a DDK-dependent and CDK-independent manner.
2. Page 25 legend for Fig 3C, "Sld3/7 recruitment reaction conducted as in Fig. 2A"; I think the reaction was carried out as in Fig 3A and analyzed as in Fig 2A.
3. Does Sld3 bind to a monomeric form of phosphorylated Mcm4 and Mcm6? Is there any difference between a single hexamer and a double hexamer of phosphorylated Mcm2-7 in the interaction with Sld3? These information, if provided, could help our understanding of regulation of replication initiation.

Referee #3:

The manuscript from Diffley and colleagues describes a detailed analysis of the first step of the activation of the eukaryotic replicative helicase, the Mcm2-7 complex. It has been known for some time that this event requires the phosphorylation of multiple Mcm2-7 subunits by the DDK kinase and previous studies had shown that, in particular, the association of Cdc45 and Sld3 with the origin required this modification. Despite this, how DDK phosphorylation of Mcm2-7 led to the recruitment of these factors has been unclear. Because deletion of the N-terminal region of the Mcm4 subunit led to the bypass of DDK function *in vivo*, it has been suggested that Mcm2-7 phosphorylation by DDK relieved an inhibitory effect of this region. In contrast to this hypothesis, the studies presented here support a model in Sld3 recognizes phosphorylated peptides directly.

Using a variety of reconstituted assays for helicase activation and replication initiation, the authors find that Sld3 is required for Cdc45 recruitment to the origin but recruitment of Sld3 and Sld7 occurs independent of Cdc45. Importantly, the authors show that the recruitment of Sld3 and Sld7 is strongly stimulated by DDK phosphorylation of Mcm2-7 and inhibited by Rad53 phosphorylation of Sld3. The authors identify regions of Sld3 that bind Cdc45 and Mcm2-7 and generate mutants that interfere with these interactions and with successful replication initiation. The authors identify a number of phosphorylated Mcm6 peptides that Sld3 can bind and show that mutations that eliminate these sites are defective for Sld3 binding in their *in vitro* assays and that combining all of the mutations leads to *in vitro* replication defects and cell lethality. Finally, the authors show that conversion of these phosphorylation sites to phosphomimetic aspartates leads to a Mcm2-7 complex that can bind to Sld3 and initiate replication *in vitro* in the absence of added DDK.

Overall this is a very nice story that provides important insights into kinase regulation of DNA replication initiation. Finding that Sld3 is connected to the action of DDK (here), Rad53 (here and previous studies) and CDK (previous studies) is an important advance. The biochemical data providing a more refined understanding of how DDK phosphorylation stimulates Sld3 and Cdc45 recruitment is also important. This is a very nice story that will be of significant interest to those interested in DNA replication, cell cycle progression and the control of multi-protein complex assembly.

Specific points:

1. The legend for Fig. EV3 is missing a description of the (B) part of the figure.
2. The experiments addressing the interaction between individual Mcm subunits and Sld3 clearly show that Mcm4 and Mcm6 are selectively precipitated ((Fig. EV3). One minor point is that they do not show that they are independently bound, since it is possible that the two proteins are still associated with one another in the precipitation as Fig. EV3A only shows that Mcm3 is not bound to

the rest of the subunits after the high salt treatment. It would be nice to use purified individual Mcm4 and Mcm6 to show that they can independently bind Sld3.

1st Revision - authors' response

07 January 2016

Referee #1:

We were pleased that this referee felt that the “...*quality of experiments is high; each experiment is well designed to draw the results and probably only few labs are able to perform this kind high quality experiments because of the requirements of many purified proteins and accumulated skills for the in vitro assays. Thus this paper is worth publishing in the EMBO Journal.*”

S/he felt there were two major points that needed addressing before publication.

1. Is it really phosphorylation specific? The authors used phosphorylated peptides and mutated Sld3 proteins for this analysis. However, it is still conceivable that the surface charge (acidophilic charges caused by phosphorylation) of the specific sequence in Mcms recruits Sld3. Although this is a difficult question, the authors may discuss the difference of affinities between phospho-mimicking mutant and phosphorylated wild type proteins.

As the reviewer notes, Sld3 appears to bind similarly well to both phosphorylated wild type MCM and MCM mutants in which serines/threonines were mutated to acidic residues (but not alanine residues), suggesting that the negative charges on the N-termini of Mcm4/6 are sufficient to recruit Sld3. Indeed, we presume this is why these mutants can bypass DDK function *in vitro* and *in vivo* and are, therefore, genuinely ‘phospho-mimicking’. We note that the Sld2 T84D mutant binds BRCT repeats 3 and 4 of Dpb11 (without phosphorylation), whilst the equivalent acidic amino acid substitutions in Sld3 do not allow Sld3 to bind BRCT repeats 1 and 2 of Dpb11. So, even within a family of phosphopeptide binding domains, sometimes the acidic amino acid substitutions work and sometimes they don't. Only with further detailed structural analysis would we be able to identify the key determinants of phosphopeptide binding by Sld3. We have added a sentence in the first paragraph of the Discussion to highlight these points.

2. There are two discrepancies between the results of this paper and previous in vivo observations. First, in vivo analysis of budding yeast showed that Sld3 associates with origins in a Cdc45-dependent manner (Kamimura et al., 2001) while the authors showed in this paper that Sld3 alone can bind to Mcm2-7 in vitro. Second, Sld7, which binds to Sld3, was reported to associate with origins in an Sld3-dependent manner (Tanaka et al., 2011). Although I do trust the results obtained in this paper (that is, Sld3 is a key protein recognizing the phosphorylated Mcms), these discrepancies can be explained if we assume some other factors in vivo, for example. Thus, the authors should discuss this possibility carefully.

We have now included additional discussion in the results section ‘DDK promotes Sld3-dependent Cdc45 recruitment’ to attempt to explain apparent discrepancies between our data and that reported in Kamimura et al., 2001. As discussed, we think there may be several possible explanations for this discrepancy.

We have included an extra figure (Fig. EV1A) showing that Sld7 (but not Sld3) binding to MCM is abolished when the salt concentration is increased in our *in vitro* assays. Thus, the phospho-MCM-binding activity of Sld7 is relatively weak, and we propose this as the reason for the differences between our results and

Tanaka et al., 2011. Consistent with this, as we show later in the paper, Sld7-dependent recruitment of an MCM-binding mutant of Sld3 cannot support DNA replication.

(Minor and miscellaneous points)

1. The authors discuss the interaction between Sld3 and Mcm2-7 in Fig.7. The published partial structure of Sld3/Sld7 and the proposal model described in the same report (Itou et al., *Acta Cryst. D71*, 1649-1656, 2015) may further support the authors' model.

We thank reviewer 1 for bringing this to our attention. We have now referred to this structure in the discussion.

2. p.5 line 4, "to be" is repeated.

We have removed this.

3. p.6, 2nd paragraph, line 3, "lanes 6 and 8" should read "lanes 4 and 8".

We have changed this.

4. Some references lack pages, please fix them.

These are now fixed

5. There are many supplementary data. If the page limit allows, I recommend showing some of them in main results (for example, peptide screenings). It is much easier for audience to follow the text.

The main text already contains seven rather dense figures, and in response to another referees comments, we have moved Fig. EV3C into the main figures, so we are reluctant to move all of the complicated peptide binding data into the main figures. We note that the expanded view figures should appear with the main text online, and readers will be able to access the additional information here and in the appendices whilst reading the manuscript.

Referee #2:

There are two major concerns.

The most important finding of this paper is that Sld3 binds specifically to DDK-phosphorylated polypeptides of Mcm4 and Mcm6. Nonetheless, contributions of DDK-phosphorylation to Sld3 binding were not clearly distinguished from DDK-independent phosphorylation at polypeptide level. In addition, the detailed results of the phosphorylated peptides of Mcm4, which seems to be more important than Mcm6, were not presented. The second problem is that the manuscript is premature in logical writing. The aim of the experiment was not properly presented at several places and the important results were not precisely described. Extensive rewriting is required. The individual comments are shown below.

1. As stated in the Abstract, the authors found that Sld3 binds specifically to DDK-phosphorylated peptides from Mcm4 and Mcm6. However, this conclusion is not readily supported by the results presented in the main figures or in the supporting results (Fig EV5). I felt difficulty in distinguishing the phosphorylated residues that interacted with Sld3 (Fig EV5A and B). For example, T66P and S69S exhibit similar intensity in EV5A but only T66 is listed in EV5B. In addition, it is not clear how DDKphosphorylation contributes to the interaction of specific polypeptides with Sld3. The complete set of the results and the criterion for the positive interaction should to be included. More importantly, the detailed results of Mcm4 polypeptides were not presented, despite of importance of Mcm4 in DDK-dependent regulation. Comprehensive presentation of the DDKdependent

and DDK-independent phosphorylation sites found in the Sld3-interacting polypeptides might be required.

We agree that the major finding in our paper is that Sld3 binds specifically to DDK-phosphorylated peptides in Mcm4 and 6, and we feel that we have established beyond reasonable doubt that this is the essential function of DDK in replication. This reviewer's main concerns are around the nature of the Sld3 binding sites and their relation to DDK phosphorylation sites in MCM subunits, rather than the fact that Sld3 binds them. Before addressing his/her specific points, I would like to comment on the complexities involved in dealing with such an enormous number of phosphorylation sites. I think the reviewer will agree that we have clearly identified some DDK-dependent phosphopeptides that bind Sld3. However, I freely acknowledge we have probably not identified all DDK phosphopeptides that bind Sld3 in a functionally relevant manner. The first problem is to identify all the DDK sites. We have performed extensive mass spectrometry on DDK-phosphorylated MCM and, despite many attempts using different proteases, etc., we (and Randell et al.) have never been able to get more than ~75% 'coverage' of the MCM subunits. Consequently, there are many potential – even likely – DDK sites we simply have no information on because they don't 'fly' in mass spec. Taken with the peptide array data, this means that we have identified some Sld3-interacting phosphopeptides which we cannot unambiguously assign as being DDK-dependent. The second problem is identifying all the phosphopeptides Sld3 can bind. Peptide arrays are very useful to screen large numbers of peptides, but they have some limitations as well. For technical reasons, we cannot synthesise peptides on membranes with more than one phospho-amino acid. Consequently, we can only use arrays to examine binding to singly phosphorylated peptides. Given Sld3 prefers binding multiply phosphorylated peptides, it is very possible we missed some DDK peptides that can bind Sld3. In addition, current technology limits the length of peptides that can be synthesised on membranes. So, steric effects on binding when the relevant phospho-site is close to the site of attachment to the membrane can be a problem. These problems can be obviated somewhat by using longer, HPLC-purified phosphopeptides attached to beads. However, the costs and labor associated with this approach make it unfeasible to analyse all ~50 sites (and combinations of sites) on Mcm4/6 this way. As a consequence, we analysed a few phosphopeptides by this method. The third problem is redundancy. We know from Randell et al. that there is a huge amount of 'redundancy' in the function of DDK sites in Mcm4. Consistent with this functional redundancy, we have shown that there is also a great deal of redundancy in the binding of Sld3 to different DDK sites in Mcm6 (Fig. EV4 and 5). It is easy to see that, with ~50 sites, the number of combinations of 2, 3, 4 etc. sites quickly escalates beyond the doable.

This is a rather long-winded way of explaining why it is virtually impossible, in a single paper like this, to have a complete analysis of DDK and Sld3 sites in MCM. I have significantly expanded our description of how we arrived at Mcm6- Δ N,11A and Mcm4-25A, and I have also included a detailed explanation of the issues described above in a new second paragraph in the Discussion, which I think should satisfy this reviewer's objections.

Specific points

For the reasons outlined above, we did not deconvolve all of the sites in Mcm4, but simply used the Randell data to design our mutant. Nonetheless, we appreciate the importance of the Mcm4 peptide data, and have thus included a new table (Fig. EV5H), detailing the Sld3-interacting phosphorylated peptides in Mcm4 identified by peptide array.

The phosphopeptide arrays consisted of peptides of overlapping sequence, as is detailed in the Materials & Methods. Only one phosphorylated peptide (not shown in Fig. EV5A) containing T66P bound to Sld3 (column 3, Fig. EV5B). Several other peptides contained this phosphorylated residue but did not bind Sld3 (as in Fig. EV5A). As described above, this may be due to presentation of the binding site on the array, or to some other local sequence context that may

play some role in Sld3 binding. To avoid confusion, we have removed part of Fig. EV5A, including the non-interacting T66P peptide.

2. Relating with the above points, some of the supporting data (Fig EV5) are better to be included in the main figures. The authors need to provide enough information for individual phosphorylation sites.

Whilst we appreciate the importance of the data in Fig. EV5 for the manuscript, inclusion of this volume of data with the main text is impractical. Hence, we include a single example of an Sld3-interacting phosphopeptide in Fig. 4E, which confirms that Sld3 is indeed a phosphopeptide binding protein. Specialist readers interested in the details of the individual phosphorylation sites can find them in the expanded view figures, which should accompany the main text in the online version.

3. I recommend to include Fig EV3 B and C instead of Fig 4B, because the latter figures demonstrated the preferential interactions of Mcm4 and Mcm6 with Sld3.

We have incorporated Fig. EV3C into the main text as Fig. 4B.

4. I think the manuscripts need to be reorganized to clarify the logical flow. For example, the second section "MCM binding is an essential function of Sld3", was difficult to follow. In page 5 line 20, the aim of the experiments should be stated before describing the results. Then, the experiments for Rad53 (Fig 2E) from page 5 line 26 to page 6 line 6 should be described later, for example, after the results of amino acid substitutions (Fig 2F and Fig 3). Furthermore, the aim of the experiment is again missing at the beginning of the third paragraph (page 6 line 13). Despite of importance of Fig 3D and 3E, the data were not precisely described. This section needs to be reorganized.

We apologise if reviewer 2 found the section ‘MCM-binding is an essential function of Sld3’ confusing. Rather than re-organising the section, which might lead to more confusion, we have re-worded and included extra explanation for portions of the text this reviewer found confusing.

5. Page 8 line 3, the aim of the experiment (Fig 5) should be stated before describing the results.

We have done this.

6. Page 8 lines 10-14, the authors need to mention the significance of severer effects of the Mcm4 mutant than the Mcm6 mutant.

We have included a sentence to this end.

7. Fig. 6 might be moved into the supporting data, because the question raised seems to be less interesting than others.

In our opinion, the experiments shown in Figure 6 are the first direct evidence that MCM phosphorylation is not required for ongoing replication fork progression, consistent with our conclusion that DDK executes its essential function by recruiting Sld3 to MCM during origin firing. Reviewer 1 agrees that this is an ‘important finding for DNA replication and checkpoint control’. Thus, we feel it appropriate to keep this figure with the main text of the article.

8. Page 10 lines 10-14, the results in Fig. 7C and 7D should be described separately and more precisely. The individual mcm6-25D and mcm4-25D mutants promoted Sld3 recruitment at a similar level with the double

mutant in the absence of DDK (Fig 7A), whereas they did not efficiently support Cdc45 or GINS loading compared with the double mutant (Fig 7C). Do these results suggest some other roles of DDK-phosphorylation on Mcm4 and Mcm6? The authors need to explain the difference. In addition, the description "at levels that approached that achieved by the wild type MCM complex with DDK" did not precisely reflect the data and should be improved.

We have included additional explanatory text for this section. Additionally, we have replaced "at levels that approached that achieved by the wild type MCM complex with DDK" with "at approximately 60% the level achieved by the wild type MCM complex with DDK".

The reviewer is correct regarding the apparent disconnect between the ability of the phospho-mimicking MCM mutants to support Sld3/7 recruitment and their ability to support DDK-independent replication (i.e. Mcm6-25D and Mcm4-25D can support similar levels of Sld3/7 recruitment but Mcm4-25D supports greater levels of replication). We have included a discussion of this in the penultimate paragraph of the section "Phospho-mimicking MCM mutants support DDK-independent replication" for clarity.

9. Because Fig 7D showed only partial information, it should be replaced with Fig EV7A.

As is shown in Figure EV6, the various phospho-mimicking MCM mutants each exhibit some defect in their ability to be loaded onto DNA. We thus had to adjust the concentration of MCM in our reactions to equalize the amount of loaded MCM between samples (Fig. EV6C). Whilst this adjustment was made for the experiment in Fig. 7D, it was not for Fig. EV7A. We thus feel Fig. 7D is the most accurate representation of the levels of DDK-independent replication observed for the Mcm4/6 phospho-mimicking mutants.

10. Page 10 lines 18-19 and Fig EV7B, suppression of cdc7-4 defect by mcm6-18D at 30°C was not convincing. Does mcm6-18D in combination with mcm4-25D show better suppression of cdc7-4 than mcm4-25D alone? Since some data are confirmation of the previous work, the clear evidence for contribution of Mcm6-phosphorylation should be provided for the novelty.

Evaluating the phenotype of the acidic substitution mutants is complicated because, in addition to suppressing loss of Cdc7, they also have defects in loading (Fig. EV6B), and combining *mcm4* and *mcm6* mutants exacerbates the loading defect. Consequently, the double mutant might actually appear to suppress less well than single mutants and might even be synthetically lethal. We don't think there is much to be learnt from this analysis, and we disagree with the reviewer regarding *mcm6-18D*; we think the partial suppression is fairly clear. Fig. EV7A shows that an Mcm6 phospho-mimicking mutant can support DDK-independent replication *in vitro*. Furthermore, a combined Mcm4/6 mutant supports more DDK-independent replication than Mcm4 25D alone. This *in vitro* data indicates that a phospho-mimicking Mcm6 mutant can support a small amount of replication even in the absence of canonical Mcm4 phosphorylation by DDK.

Minor comments

1. Page 5 lines 5-6, "Sld3 origin association clearly precedes Cdc45 (Nakajima et al., 2002) in fission yeast"; the report by Yabuuchi et al (2006) should be the appropriate reference, because it showed that Sld3 but not Cdc45 is loaded onto origin in a DDK-dependent and CDK-independent manner.

We apologise for this omission, and have now included the appropriate reference.

2. Page 25 legend for Fig 3C, "Sld3/7 recruitment reaction conducted as in Fig. 2A"; I think the reaction was carried out as in Fig 3A and analyzed as in Fig 2A.

The experiment in Fig. 3C was conducted as in Fig. 2A.

3. Does Sld3 bind to a monomeric form of phosphorylated Mcm4 and Mcm6? Is there any difference between a single hexamer and a double hexamer of phosphorylated Mcm2-7 in the interaction with Sld3? These information, if provided, could help our understanding of regulation of replication initiation.

Data in Figures 4 and EV3 show that Sld3 can bind to monomeric Mcm4 and Mcm6, which have been previously phosphorylated as part of the MCM double hexamer. Biochemical evidence from the Bell and Speck labs has established that DDK preferentially phosphorylates MCM loaded onto DNA over other forms of the helicase. This was the reason for establishing the complicated protocol used in this study to examine interactions of Sld3/7 with individual Mcm subunits. Whilst it would potentially be interesting to see if Sld3/7 can bind individual MCM hexamers, we cannot do this because DDK doesn't phosphorylate these single hexamers.

Referee #3:

We were pleased this reviewer was quite positive about our work: "Overall this is a very nice story that provides important insights into kinase regulation of DNA replication initiation. Finding that Sld3 is connected to the action of DDK (here), Rad53 (here and previous studies) and CDK (previous studies) is an important advance. The biochemical data providing a more refined understanding of how DDK phosphorylation stimulates Sld3 and Cdc45 recruitment is also important. This is a very nice story that will be of significant interest to those interested in DNA replication, cell cycle progression and the control of multi-protein complex assembly." S/he raised several specific points.

Specific points:

1. The legend for Fig. EV3 is missing a description of the (B) part of the figure.

Fig. EV3 has now been altered in line with suggestions from reviewer 2. The figure legend has been changed accordingly.

2. The experiments addressing the interaction between individual Mcm subunits and Sld3 clearly show that Mcm4 and Mcm6 are selectively precipitated ((Fig. EV3). One minor point is that they do not show that they are independently bound, since it is possible that the two proteins are still associated with one another in the precipitation as Fig. EV3A only shows that Mcm3 is not bound to the rest of the subunits after the high salt treatment. It would be nice to use purified individual Mcm4 and Mcm6 to show that they can independently bind Sld3.

We have been unable to reconstitute efficient phosphorylation of individual Mcm subunits with purified DDK in vitro, making such experiments difficult. However, we have isolated individual Mcm4 and Mcm6 mutants that do not interact with Sld3/7. These mutants have no impact on the amount of the other Mcm subunit precipitated by Sld3/7 (Fig. 4F). Thus, we conclude that Mcm4 and Mcm6 are each independently bound to Sld3/7 in our experiments.

2nd Editorial Decision

14 January 2016

Thank you for submitting your revised manuscript for our consideration. It has now been assessed once more by one of the original referees, whose comments are copied below. I am happy to say that this referee is fully satisfied with the revisions, and we shall therefore be able to proceed with formal acceptance of the paper, pending addressing of a few remaining editorial points as follows:

* We still require a completed Author Checklist file from you (see below for download link)

* For the Expanded View figures, which are currently somewhat blurry/pixelated, please upload new higher-quality/higher-resolution versions, and please also provide us with the respective source data to facilitate our assessment. Please also make sure to upload each EV figure individually.

* in addition, I would also encourage you to submit figure source data for all or at least for key gels/blots/autoradiographs in the main figures, in order to make the primary data behind the various blot/gel panels more accessible and more directly represented. We would ask for a single PDF/JPG/GIF file per figure comprising the original, uncropped and unprocessed scans of all gel/blot panels used in the respective figures. These should be labelled with the appropriate figure/panel number, and should have molecular weight markers; further annotation would clearly be useful but is not essential. These files can be uploaded upon resubmission selecting "Figure Source Data" as object type, and they would be linked as such to the respective figures in the online publication of your article.

* I would suggest to revise the manuscript title to make it more immediately accessible for a broad readership, including adding reference to 'replication'. For example, how about "Phosphopeptide binding by Sld3 links Dbf4-dependent kinase to MCM helicase activation during replication" or "Phosphopeptide binding by Sld3 links Dbf4-dependent kinase to MCM replicative helicase activation"?

REFeree COMMENTS

Referee #2:

I think that the author's responses and the changes made in the revised version of the manuscript are satisfactory.

2nd Revision - authors' response

02 February 2016

Thank you for your positive final decision. I have uploaded the completed author's checklist, a revised MS with new title and higher resolution versions of all the figures. We felt there was little use in submitting source files. All of our experiments were performed with purified proteins so there are no issues with antibody cross-reactivity or proteolysis of proteins. We routinely cut membrane slices from blots after transfer across ranges of molecular weights so we can look at multiple proteins in a single experiment. So, there are no full gel-sized blots. Films from the immunoblots were scanned and the only manipulation was to adjust levels, making sure everything is in the linear range, before assembling into a final figure. We hope this is acceptable.

Corresponding Author Name: John F.X. Dillley
 Journal Submitted to: The EMBO Journal
 Manuscript Number: EMBOJ-2015-93552R